# CfDNA Measurement as a Diagnostic Tool for the Detection of Brain Somatic Mutations in Refractory Epilepsy

**DOI:** 10.3390/ijms23094879

**Published:** 2022-04-28

**Authors:** Sonia Mayo, Irene Gómez-Manjón, Francisco Javier Fernández-Martínez, Ana Camacho, Francisco Martínez, Julián Benito-León

**Affiliations:** 1Genetics and Inheritance Research Group, Instituto de Investigación Sanitaria Hospital 12 de Octubre (imas12), 28041 Madrid, Spain; soniamayodeandres@gmail.com (S.M.); irenegomezmanjon@hotmail.com (I.G.-M.); ffernandezm@salud.madrid.org (F.J.F.-M.); 2Department of Genetics, Hospital Universitario 12 de Octubre, 28041 Madrid, Spain; 3Division of Pediatric Neurology, Department of Neurology, Hospital Universitario 12 de Octubre, 28041 Madrid, Spain; acamachosalas@yahoo.es; 4Department of Medicine, Universidad Complutense de Madrid, 28040 Madrid, Spain; 5Traslational Research in Genetics, Instituto de Investigación Sanitaria La Fe (IIS La Fe), 46026 Valencia, Spain; martinez_fracas@gva.es; 6Genetics Unit, Hospital Universitario y Politécnico La Fe, 46026 Valencia, Spain; 7Department of Neurology, Hospital Universitario 12 de Octubre, 28041 Madrid, Spain; 8Centro de Investigación Biomédica en Red sobre Enfermedades Neurodegenerativas (CIBERNED), 28031 Madrid, Spain

**Keywords:** cell-free DNA, somatic mutations, refractory epilepsy

## Abstract

Epilepsy is a neurological disorder that affects more than 50 million people. Its etiology is unknown in approximately 60% of cases, although the existence of a genetic factor is estimated in about 75% of these individuals. Hundreds of genes involved in epilepsy are known, and their number is increasing progressively, especially with next-generation sequencing techniques. However, there are still many cases in which the results of these molecular studies do not fully explain the phenotype of the patients. Somatic mutations specific to brain tissue could contribute to the phenotypic spectrum of epilepsy. Undetectable in the genomic DNA of blood cells, these alterations can be identified in cell-free DNA (cfDNA). We aim to review the current literature regarding the detection of somatic variants in cfDNA to diagnose refractory epilepsy, highlighting novel research directions and suggesting further studies.

## 1. Introduction

Epilepsy is a chronic neurological disorder affecting more than 50 million people worldwide [1]. Epileptic patients, especially drug-resistant ones, have increased risks of premature death, injuries, psychosocial dysfunction, and reduced health-related quality of life [2]. Moreover, approximately 30% of these patients present refractory epilepsy (RE) [3], defined as “failure of adequate trials of two tolerated and appropriately chosen and used anti-seizure medication schedules (whether as monotherapies or in combination) to achieve sustained seizure freedom” [4].

The etiology of epileptic disorders is unknown in an estimated 60% of patients [5]. Some genetic component is suspected in up to 70–80% of cases [6]. Genetic alterations responsible for epilepsy are very heterogeneous, spanning from chromosomal rearrangements and copy number variants (CNVs) to single nucleotide variants (SNVs) [7,8]. It has been recognized for a long time that epilepsy in infants and children (often severe and intractable) occurs together with developmental abnormalities and intellectual disability (ID) [9]. Epilepsy is among the most common findings associated with chromosomal aberrations, particularly those involving chromosomal imbalance [10]. Initially, rearrangements associated with epilepsy, ID, and congenital anomalies were detected by karyotyping [11]. In those cases, advances in molecular cytogenetic techniques have led to the replacement of karyotyping by microarrays, which are capable of detecting CNVs smaller than 5 Mb [12]. Pathogenic deletions and duplications have been identified in complex genetic disorders such as ID or autism spectrum disorders (ASD) [13,14]. CNVs are also an important molecular cause of RE, such as in most epileptic encephalopathies (EE), with up to 8% of cases carrying CNVs that are causative or potentially contributing to the pathology [15]. The pathogenicity of CNVs in complex diseases is well known, such as in tuberous sclerosis complex, which may present with RE and ID [16]. However, genetic variants are currently the most frequent cause of childhood EEs [7]. In 2001, Claes et al. [17] described *de novo* mutations in *SCN1A* as a cause of Dravet syndrome. Since then, hundreds of genes involved in epilepsy have been discovered, and their number is progressively increasing [18,19]. The use of screening techniques based on next-generation sequencing (NGS) has allowed the identification of new pathological genes in 7–9% of the studied cases, obtaining a molecular diagnosis in more than 38% of patients [20]. The genetic diagnosis based on NGS approaches can improve treatment efficacy and reduce hospitalization, especially in children with RE [21]. However, there are still some cases in which the results of the molecular studies do not fully explain the clinical phenotype of the patients.

Epilepsy has a broad phenotypic and genotypic heterogeneity; different mutations in the same gene can lead to diverse syndromes and phenotypes, while different genes can cause the same epileptic syndrome. For example, mutations in *KCNQ2* cause benign familial neonatal seizures. However, they are also associated with severe neonatal EE characterized by tonic seizures and developmental impairment, sometimes presenting as Ohtahara syndrome [22]. Furthermore, patients with RE can have different genetic alterations at diverse *loci* with a combined effect that explains this highly variable phenotype. In this sense, it is well known that in individual carriers of pathogenic variants at *SCN1A*, the presence of a second mutation in a modulator gene, such as *SCN9A*, *CACNA1A*, *POLG*, or *CACNB4*, can modify their phenotype, causing a wide clinical spectrum [23]. In addition, complex genetic pathologies associated with RE also present ID and/or ASD, such as in neuronal migration disorders [24]. Therefore, the new screening techniques based on NGS are fundamental to evaluating all the possible genetic and genomic variants involved in the molecular etiology of epilepsy. (See Appendix A, Table A1, for details about some NGS-based methods).

Sequencing is most commonly applied to lymphocyte-derived DNA to search for pathogenic germline variants [9]. However, disease-causing mutations can also occur during the mitotic cell divisions, leading to mosaic individuals with only a subset of their cells harboring the mutation [25]. This somatic variant may be tissue-specific. Somatic mutation involving the brain can occur at any time in life [26]. Recent studies have established a role for somatic mutations in several neurological diseases affecting children, such as epilepsy disorders [27]. Specifically, analysis of brain tissue from patients undergoing neurosurgery for refractory epilepsy has allowed the identification of somatic variants undetectable in DNA from peripheral blood [26]. However, cell-free DNA (cfDNA), released after cell death, circulates in different body fluids and can be analyzed with specific strategies.

Different literature reviews have focused on the relation of somatic mutations in epilepsy [26] or the role of plasmatic cfDNA as a biomarker in neurological disorders [28]. cfDNA from cerebrospinal fluid (CSF) has been investigated to target detection of somatic mutations in epilepsy. However, no study has pointed out the possibility of brain–cfDNA analysis in plasma for somatic variant detection in RE, which would increase the diagnostic yield of this disease with a minimally invasive procedure. In this review, state-of-the-art approaches regarding somatic brain variants in RE are summarized, and the utility of cfDNA to detect those types of variants is discussed.

## 2. Discussion

### 2.1. Somatic Variants in the Brain and Refractory Epilepsies

Genetic changes occur during development and are accumulated during an individual’s life. As an example, monozygotic twins discordant for an *SCN1A* mutation have been described (a Dravet syndrome patient, carrier of the variant c.664C>T; p.(Arg222*) vs. her twin, without detected mutation of DNA from lymphocytes, hair, buccal cells, skin fibroblasts, and cell lines derived from the olfactory neuroepithelium, was only affected by two simple febrile seizures before four years of age) [29]. Although most neurons persist without replacement once formed during early development, post-mitotic mutation may alter brain function and contribute to neurological disease [26]. The long life span of individual neurons and the direct relationship between neural circuits and behavior suggest that somatic mutations in small populations of neurons would be enough to affect individual neurodevelopment significantly [30]. Some studies have suggested that the brain may have widespread somatic mutations in its development [25,31,32]. On the other hand, a Brain Somatic Mosaicism Network has been developed to examine large numbers of neurons from neurotypical controls alongside matched individuals with different neurological disorders, including focal cortical dysplasia (FCD) in order to evaluate the implication of common brain somatic mutations in neurological disease (https://bsmn.synapse.org/index.html; accessed on 21 March 2022) [30]. Therefore, brain somatic variants might explain and be the cause of previously undiagnosed RE. Most of these variants are undetectable in blood DNA. They have been found by analyzing brain tissue from surgical or autopsy specimens, which are not available for the most common epilepsies and other neurological and neurodevelopmental disorders [33].

Many studies have identified somatic pathogenic variants specific to brain tissue from RE patients by comparing variations in DNA from pathological tissues with DNA from leukocytes from the same individuals (Table 1).

In particular, different genes encoding components of the PI3K-AKT3-mTOR pathway have been linked to conditions associated with RE, such as hemimegalencephaly (HME) or FCD. In 2012, Poduri et al. [37] analyzed brain tissue from surgical resection of eight patients with HME, a brain malformation associated with epilepsy. In two cases, they identified partial trisomy of 1q, including *AKT3*. In a third case, a somatic missense variant in this gene indicated that the somatic activation of *AKT3* was responsible for this brain malformation [37]. The same year, Lee et al. [38] identified *de novo* somatic mutations in *PIK3CA*, *AKT3*, and *MTOR* genes in six individuals affected by HME from 20 cases. In 2015, eight somatic brain activating mutations in *MTOR* were also associated with FCD type II (FCDII), suggesting mTOR as a treatment target for ER [48]. D’Gama et al. [39], with a custom panel specific for mTOR pathway genes, identified 14 somatic pathogenic variants, of which seven were not found in blood from patients with FCD and HME. Similarly, Baldassari et al. [40] detected 34 somatic variants in mTOR pathway genes in 80 children with drug-resistant epilepsy and genetic malformations of cortical development (mild MCD, FCD, or HME) [40]. Avansini et al. [34] found three somatic variants in two patients with FCDIIb, focusing on 60 genes of the mTOR pathway. Lee et al. [50] analyzed 20 operated patients with refractory focal epilepsy and bottom-of-sulcus dysplasia, screening for somatic variants of 331 genes and detecting three different pathogenic variants in *MTOR* in six individuals [50]. Besides, Zhang et al. [46] detected seven somatic (probably pathogenic) variants in six from 17 children with FCDII. Interestingly, five of the seven identified genes (*IRS1*, *RAB6B*, *RALA*, *HTR6*, and *ZNF337*) had not been previously associated with cortical malformation. An in vitro functional study demonstrated that the *IRS1* variant led to mTOR hyperactivation [46].

In 2016, Hildebrand et al. [35] found somatic variants in 14 patients with hypothalamic hamartoma epilepsy affecting the Sonic hedgehog (Shh) pathway (*PRKACA*, *GLI3*, *CREBBP*, and *WNT11*, among other genes) from 38 individuals. Remarkably, six loss-of-heterozygosity (LOH) were reported in 43% of these variants. For instance, LOH without a deletion can be due to uniparental disomy, a pathogenic mechanism associated with some imprinting syndromes. In addition, one patient was found to be a carrier of somatic CNVs [35]. The Shh pathway plays an important role in neural development, and brain cancer cells use this mechanism to resist chemotherapeutic drugs [59].

In 2018, Winawer et al. [56] identified five somatic (probably pathogenic) variants in *SLC35A2* from a cohort of 58 individuals with RE (18 with non-lesional focal epilepsy and 38 with focal malformations of cortical development). The same year, Sim et al. [55] also identified in six patients with Lennox-Gastaut syndrome different somatic pathogenic variants in *SLC35A2.* Moreover, Bonduelle et al. [54] identified nine somatic pathogenic variants in *SLC35A2* from 20 children with mild malformation of cortical development with oligodendroglial hyperplasia in epilepsy (MOGHE). Additionally, 17 more cases with pathogenic *SLC35A2* variants from an international consortium were included [54]. After a histopathological evaluation, these cases, initially classified as malformations of cortical development (MCD), were reclassified as MOGHE. Indeed, Bonduelle et al. [54] proposed that mosaic *SLC35A2* variants, which likely occurred in a neuroglial progenitor cell during brain development, could be a genetic marker for MOGHE. *SLC35A2* (MIM * 314375) encodes a member of the nucleotide-sugar transporter family. Mutations in this gene cause a congenital disorder of glycosylation type IIm (CDG2M), characterized by severe or profound global developmental delay and early epileptic encephalopathy, among other clinical features (MIM #300896).

In 2019, Sim et al. [41] screened for somatic mutation in resected brain tissue from 232 ER patients. They focused on 28 epilepsy-related genes, detecting 51 somatic variants, of which 26 were pathogenic or probably pathogenic, as classified by the American College of Medical Genetics [41]. Niestroj et al. [43] identified 13 somatic variants, six pathogenic and seven probably pathogenic, in 54 individuals with epilepsy-associated brain lesions [43]. Blümcke et al. [42] detected five somatic (probably pathogenic) variants in four genes in 22 patients within the spectrum of focal cortical dysplasia. Furthermore, target approaches identified specific somatic brain variants in some pathologies, such as the forme fruste of Sturge–Weber syndrome (SWS) [45]. Different case reports showed similar results [36,47,49,51,52,53,57,58].

In summary (Table 1), most of the somatic variants are detected in genes from the mechanistic target of the rapamycin (mTOR) pathway (Figure 1). This pathway regulates various brain functions, from brain development to degeneration [60]. Moreover, germline and somatic mutations activating the mTOR pathway are responsible for RE [61,62,63]. Furthermore, somatic variants have also been identified in other important pathways involved in neuronal development and drug response, such as the Shh pathway. In addition, specific brain somatic variants could help define the phenotypic spectrum associated with particular genes, such as in *SLC35A2*.

### 2.2. Somatic Variants in CSF CfDNA and Refractory Epilepsy

#### 2.2.1. Cell-Free DNA

CfDNA is a mixture of extracellular nucleic acid fragments from cell necrosis, apoptosis, and active DNA release [64]. The presence of cfDNA in human plasma was first described in 1948 [65]. Since then, cfDNA has been detected in other biological fluids such as CSF [66]. The half-life of cfDNA is estimated at around an hour, disappearing after one or two days [67,68]. It has been widely analyzed as a biomarker for diagnosis, prognosis, and treatment monitoring in cancer, known as “liquid biopsy” [69]. It has also been used to study the fetal genetic complement, using fetal cfDNA from trophoblast apoptosis found in maternal plasma [70].

CSF is a source of circulating tumor DNA; cfDNA is released upon tumor cell death. This DNA is a potentially powerful biomarker for diagnosing and characterizing central nervous system tumors (CNS), such as gliomas [44,66,71,72]. Moreover, tumor-derived cfDNA in CSF samples can be used to monitor tumor progression and response to therapy [72].

#### 2.2.2. Somatic Mutations in CSF and Epilepsy

Seizures cause brain damage leading to neuronal death, especially when prolonged and repetitive, such as in ER [73]. It has been demonstrated that cfDNA can be reliably detected in CSF, with enough to perform targeted assays in epilepsy [33]. cfDNA concentration is higher in epileptic patients than in controls, which is compatible with increased apoptosis of brain cells due to seizures. This leads to the shedding of more cfDNA into CSF [33].

Brain somatic variants can be detected in the CSF-derived cfDNA in RE [33,44]. Kim et al. [44] performed a targeted analysis (ddPCR assays) in CSF from 12 patients with RE and known mosaic pathogenic variants previously identified in genomic desoxyribonucleic acid (gDNA) from brain tissue. However, only three of these alterations were identified (Table 1) [44]. Ye et al. [33] also detected three somatic variants by targeted analysis (ddPCR assays) in three patients with drug-resistant focal epilepsy. The somatic mutation was previously known in two of them, but the last one was first identified in CSF cfDNA and later confirmed in brain tissue (Table 1) [33]. Therefore, liquid biopsies’ clinical and diagnostic utility for patients with intractable epilepsy is still under investigation [44]. Until now, cfDNA from CSF has been limited to targeted assays, and technical improvement is still required before it can be used for genetic screening in epileptic patients with untargeted methods such as WGS.

### 2.3. Future Perspectives: Somatic Brain Variants Detection in Plasma CfDNA

While CSF extraction requires a lumbar puncture, an invasive procedure that might have side effects and be associated with pain and complications, blood extraction is a regular procedure nowadays that can be carried out worldwide. Therefore, the plasma would be a more suitable sample to develop diagnostic protocols in RE.

#### 2.3.1. Blood-Brain Barrier Integrity and Epitoy

To detect cerebral cfDNA in plasma, those molecules might pass through the blood-brain barrier (BBB), a dynamic and complex system that separates the brain from the blood. The integrity of the BBB is crucial for normal neuronal functioning, and alterations in functional and structural properties of this barrier are closely interrelated with the occurrence of a wide variety of CNS disorders, including epilepsy [74,75,76,77,78]. BBB disruption can directly induce seizure activity and exacerbate epileptogenesis [75,79]. However, alterations in neuronal activity have been reported to affect BBB integrity [74,80], so the relationship between epilepsy and BBB breakdown is bidirectional.

Blood-brain barrier dysfunction is observed within the first hour of status epilepticus and in epileptogenic brain regions, which may last for months [81]. Therefore, at least during this time window, cfDNA could reach the bloodstream.

#### 2.3.2. Requirements for Specific Brain CfDNA Measurement in Plasma from Patients with Refractory Epileptic

Specific differential methylation marks in plasma cfDNA have been identified in neurodegenerative diseases such as amyotrophic lateral sclerosis and multiple sclerosis [82,83]. Neuron-cfDNA is significantly elevated in response to mild trauma (wave exposure in training exercises with explosives) [84]. In addition, specific brain DNA signatures have been detected even when diluted 1:1000 in lymphocyte DNA [82]. Therefore, it has been proposed that cfDNA derived from dying CNS cells might cross the altered BBB and be isolated from peripheral blood in neurodegenerative disorders [83]. Considering this, plasma cfDNA could reflect neuronal damage in epilepsy.

Only two studies from the same research group have evaluated serum cfDNA as a biomarker in refractory focal epilepsy. Both studies suggested that cfDNA might be associated with the inflammatory and neurodegenerative process that affects the CNS in patients with RE. However, the results were different. In 2013, Liimatainen et al. [85] showed that cfDNA levels increased in most patients with refractory focal epilepsy, without association with gender, seizure type, epilepsy type, duration of epilepsy, or seizure frequency. On the other hand, Alapirtti et al. [86] showed that baseline concentrations of cfDNA were dependent on the epilepsy syndrome, even being significantly lower in patients with extratemporal lobe epilepsy (XTLE) than in healthy individuals.

In contrast, there were no significant differences between patients with temporal lobe epilepsy (TLE) and healthy controls [86]. Discrepancies might be due to the distinct sample size in both series (167 patients with focal epilepsy vs. 51 patients divided into 23 TLE, 24 XTLE, and four IGE). Alapirtti et al. [86] considered baseline samples (those collected within the 24 h previous to the first unequivocally verified seizure). However, as mentioned before, the half-life of cfDNA is around an hour, disappearing after two days [67,68]. Therefore, it would be more appropriate to compare the cfDNA concentration of samples collected during the first hour after the seizure, corrected by its baseline. In any case, neuron-specific methylation patterns should be determined to measure brain-cfDNA specifically. In this sense, a rapid and simple protocol based on ddPCR has recently been defined for tissue-specific methylation patterns of plasma cfDNA [87].

On the other hand, some molecular alterations associated with neurological tumors have already been detected in plasma cfDNA, such as the number *of MYCN* copies in neuroblastoma [88] or specific somatic mutations in glioblastoma (*EGFR* [89] and *IDH1* [90]). Therefore, at least theoretically, somatic brain mutations could also be detected in plasma cfDNA from patients with RE. However, no study has yet been published in this regard.

#### 2.3.3. Treatment

Identifying somatic brain mutations is relevant for genetic diagnoses and potential targeted therapies for many patients with epilepsy [26]. In this sense, Ko et al. [91] have evaluated the efficacy of the ketogenic diet (KD) for pediatric epilepsy in cases with germline or somatic mutations in the mTOR pathway. The difference between both groups was not statistically significant, although the sample size was very small (25 patients). Therefore, further studies in this direction are required.

On the other hand, drug resistance in epilepsy is likely to be multifactorial, and the molecular mechanisms underlying it are poorly understood. However, the transporter hypothesis is one of the most cited and accepted theories attempting to explain the neurobiological basis of multidrug resistance epilepsy [92]. The detection of the alleged and proposed pathogenic somatic variants specific to brain tissue in these transporters could be used to propose new target treatments.

## 3. Conclusions

Although a genetic factor is estimated to occur in more than 70% of epileptic patients, the etiology of RE remains unknown in more than half of all cases. Somatic mutations in tissue with such limited access as the brain could contribute to explaining these numbers. Identifying different somatic variants in DNA from epilepsy surgery specimens in recent years supports this hypothesis. In addition, it has been demonstrated that brain damage occurs as a consequence of RE in those patients. Moreover, BBB permeability could be responsible for and a result of these diseases. Neuronal-cfDNA might cross the altered BBB, and hence, somatic epilepsy mutation could be detected in plasma cfDNA. High throughput techniques might allow the screening for novel somatic mutation-specific RE in plasma cfDNA (see Appendix A, Table A1, for details about different NGS methods for detection of the ctDNA). These techniques could increase the diagnostic yield of RE with a minimally invasive procedure, enhancing the knowledge of its pathological mechanisms and improving its treatment.

## Figures and Tables

**Figure 1 ijms-23-04879-f001:**
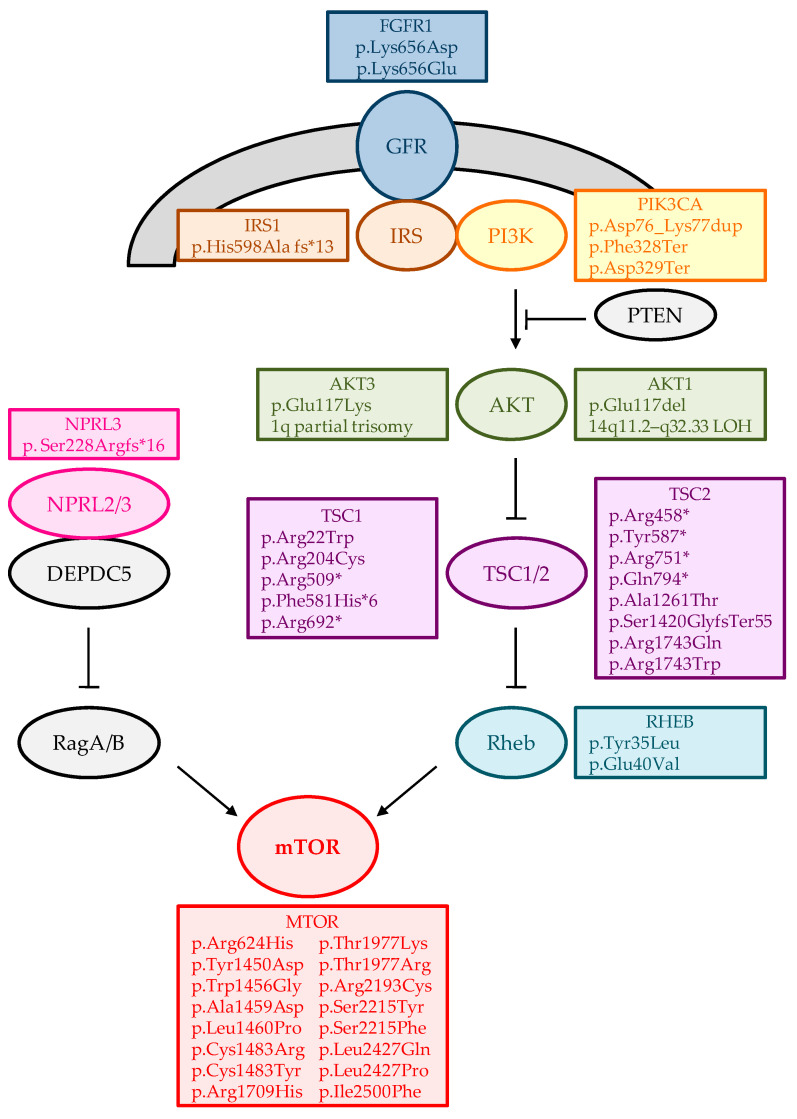
Pathogenic somatic mutations were identified in the mTOR pathway. Schematic representation of this pathway’s main proteins/complex is shown as ovals, while the somatic variants in these genes are listed in the rectangles.

**Table 1 ijms-23-04879-t001:** Review of brain somatic pathogenic variants detected in refractory-epileptic patients.

Gene/Loci	Variant	Sample	Refractory Epilepsy	N	References
*AKT1* (NM_005163)	c.349_351del; p.Glu117del	RB	FCD Iib	1 +	[34]
*BMP4*, *AKT1*	(chr14:24,419,118–106,072,470) LOH	RB	HHE	1	[35]
*AKT3*	1q21.1-q44 trisomy	RB	HME	1	[36]
*AKT3*	1q partial trisomy	RB	HME	1	[37]
*AKT3*	1q partial trisomy	RB	HME	1	[37]
*AKT3* (NM_001206729)	c.49C>T; p.(Glu17Lys)	RB	HME/FCD Iia	8	[37,38,39,40,41,42]
*BRAF* (NM_004333)	c.1799T>A; p.(Val600Glu)	RB	GG	14	[33,41,43,44]
*BRAF* (NM_004333)	c.1518_1526dup	RB	GG	1	[43]
*CREBBP*	(chr16:0–31,543,619) LOH	RB	HHE	2	[35]
*DNMT3A* (NM_175629)	c.2141C>G; p.(Ser714Cys)	RB	DNT	1	[43]
*FGFR1* (NM_023110)	c.1966_1968delinsGAC; p.(Lys656Asp)	RB	DNT	1	[43]
*FGFR1* (NM_023110)	c.1966A>G; p.(Lys656Glu)	RB	DNT	1	[43]
*GLI2*, *IHH*, *LRP2*, *STK36*, *WNT10A*, *WNT6*	(chr2:103,856,408–243,199,373) LOH	RB	HHE	1	[35]
*GLI3*(NM_000168)	c.2071C>T; p.(Gln691Ter)	RB	HHE	1	[35]
*GLI3*(NM_000168)	c.2989dupG; p.(Ala997GlyfsTer87)	RB	HHE	1	[35]
*GLI3*(NM_000168)	c.3172C>T; p.(Arg1058Ter)	RB	HHE	1	[35]
*GLI3*(NM_000168)	c.3442C>T; p.(Gln1148Ter)	RB	HHE	1	[35]
*SHH*, *SMO*, *WNT16*, *WNT2*	(chr7:58,814,064–159,138,663) LOH	RB	HHE	1	[35]
*GLI3*, *SHH*, *SMO*, *WNT16*, *WNT2*	(chr7:986,211–60,069,242;58,814,064–159,138,663) CNVs	RB	HHE	1	[35]
*GNAQ* (NM_002072)	c.548G>A; p.(Arg183Gln)	RB	ffSWS	4	[45]
*HTR6* (NM_000871)	c.G469A; p.(Ala157Thr)	RB	FCD Iib	1	[46]
*IRS1* (NM_005544)	c.1791dupG; p.(His598Ala fsTer13)	RB	FCD Iib	1 *	[46]
*KCNH1*(NM_172362)	c.2138T>A; p.(Val713Glu)	RB	FCD Iib	1	[47]
*KRAS* (NM_004985)	c.40; G>A; p.(Val14Ile)	RB	GG and HS	1	[43]
*LIS* (*PAFAH1B1*) (NM_000430)	c.190A>T; p.(Lys64Ter)	CSF	Subcortical band heterotopia	1	[33]
*MTOR* (NM_004958)	c.1871G>A; p.(Arg624His)	RB	FCD Iia	1	[48]
*MTOR* (NM_004958)	c.4348T>G; p.(Tyr1450Asp)	RB	FCD Iib	1	[48]
*MTOR* (NM_004958)	c.4366T>G; p.(Trp1456Gly)	RB	FCD Iib	2	[41,49]
*MTOR* (NM_004958)	c.4376C>A; p.(Ala1459Asp)	RB	FCD Iia/FCD Iib	5	[40,50,51]
*MTOR* (NM_004958)	c.4379T>C; p.(Leu1460Pro)	RB	FCD Iia/FCD Iib	4	[34,40,41]
*MTOR* (NM_004958)	c.4447T>C; p.(Cys1483Arg)	RB	FCD Iib	2	[41,48]
*MTOR* (NM_004958)	c.4448G>A; p.(Cys1483Tyr)	RB	HME/FCD Iib	2	[38,41]
*MTOR* (NM_004958)	c.5126G>A; p.(Arg1709His)	RB	FCD Iia	1	[48]
*MTOR* (NM_004958)	c.5930C>A; p.(Thr1977Lys)	RB	FCD Iib	8	[40,42,43,46,48,50]
*MTOR* (NM_004958)	c.5930C>G; p.(Thr1977Arg)	RB	HME/FCD	2	[39]
*MTOR* (NM_004958)	c.6577C>T; p.(Arg2193Cys)	RB	FCD Iia	1	[48]
*MTOR* (NM_004958)	c.6644C>A; p.(Ser2215Tyr)	RB	FCD Iia/FCD Iib	9	[40,41,42,50]
*MTOR* (NM_004958)	c.6644C>T; p.(Ser2215Phe)	RB	HME/FCD Iia/FCD Iib/Polymicrogyria/SKS	16	[39,40,41,48,52]
*MTOR* (NM_004958)	c.7280T>A; p.(Leu2427Gln)	RB	FCD Iia/FCD Iib	4	[41,48]
*MTOR* (NM_004958)	c.7280T>C; p.(Leu2427Pro)	RB	FCD Iia	2	[41]
*MTOR* (NM_004958)	c.7498A>T; p.(Ile2500Phe)	RB	FCD Iia	1	[40]
*NF1* (NM_000267)	c.2674del; p.(Ser892AlafsTer10)	RB	HS	1	[43]
*NPRL3* (NM_001077350)	c.682_683dup; p.(Ser228ArgfsTer16)	RB	FCD Iia	1	[43]
*PIK3CA*(NM_006218)	c.1624G>A; p.(Glu542Lys)	RB	HME/FCD Iia	3	[39,40]
*PIK3CA*(NM_006218)	c.1633G>A; p.(Glu545Lys)	RB/CSF	HME	6	[38,41,44]
*PIK3CA*(NM_006218)	c.3140A>G; p.(His1047Arg)	RB	HME/FCD Iia	2	[40]
*PRKACA*(NM_002730)	c.226-231dup; p.(Asp76_Lys77dup)	RB	HHE	1	[35]
*PRKACA*(NM_002730)	c.983_984delTT; p.(Phe328Ter)	RB	HHE	1	[35]
*PRKACA*(NM_002730)	c.984dupT; p.(Asp329Ter)	RB	HHE	1	[35]
*RAB6B* (NM_016577)	c.C383T; p.(Thr128Met)	RB	FCD Iia	1	[46]
*RALA* (NM_005402)	c.G482A; p.(Arg161Gln)	RB	FCD Iib	1	[46]
*RHEB* (NM_005614)	c.[105C>A,104A>T]; p.(Tyr35Leu)	RB	HME/FCD Iib	1	[40]
*RHEB* (NM_005614)	c.119A>T; p.(Glu40Val)	RB	HME/FCD Iib	2	[40,53]
*SLC35A2* (NM_005660)	c.935C>T; p.(Ser312Phe)	RB	MOGHE	1	[54]
*SLC35A2* (NM_005660)	c.112_116delinsTGGTGGTCCAGAATG; p.(Ile38TrpfsTer59)	RB	MOGHE	1	[54]
*SLC35A2* (NM_005660)	c.206C>T; p.(Thr69Ile)	RB	MOGHE	1	[54]
*SLC35A2* (NM_005660)	c.275-1G>T	RB	LGS/MOGHE	1	[54,55]
*SLC35A2* (NM_005660)	c.335_339dupCGCTC; p.(Lys114ArgfsTer32)	RB	MOGHE	1	[54]
*SLC35A2* (NM_005660)	c.359_360delTC; p.(Leu120HisfsTer7)	RB	MOGHE	2	[54]
*SLC35A2* (NM_005660)	c.359T>C; p.(Leu120Pro)	RB	MOGHE	1	[41,54]
*SLC35A2* (NM_005660)	c.385C>T; p.(Gln129Ter)	RB	MOGHE	1	[54]
*SLC35A2* (NM_005660)	c.502G>A; p.(Gln168Ter)	RB/CSF	LGS/MOGHE	1	[44,54,55]
*SLC35A2* (NM_005660)	c.553C>T; p.(Gln185Ter)	RB	LGS/MOGHE	2	[54,55]
*SLC35A2* (NM_005660)	c.569_572delGAGG; p.(Gly190AlafsTer158)	RB	MOGHE	1	[54]
*SLC35A2* (NM_005660)	c.580_616dupCCACTGGATCAGAACCCTGGGGCAGGCCTGGCAGCCG; p.(Val206AlafsTer28)	RB	MOGHE	1	[54]
*SLC35A2* (NM_005660)	c.589C>T; p.(Gln197Ter)	RB	LGS/MOGHE	1	[54,55]
*SLC35A2* (NM_005660)	c.603_606dupAGGC; p.(Leu203ArgfsTer20)	RB	MOGHE	1	[54]
*SLC35A2* (NM_005660)	c.634_635delTC; p.(Ser212LeufsTer9)	RB	mMCD/MOGHE/NLFE/WS	3	[40,54,56,57]
*SLC35A2* (NM_005660)	c.671T>C; p.(Leu224Pro)	RB	MOGHE	1	[41,54]
*SLC35A2* (NM_005660)	c.703A>C; p.(Asn235His)	RB	LGS/MOGHE	1	[54,55]
*SLC35A2* (NM_005660)	c.760G>T; p.(Glu254Ter)	RB	LGS/MOGHE	1	[54,55]
*SLC35A2* (NM_005660)	c.801C>G; p.(Tyr267Ter)	RB	mMCD/MOGHE	1	[40,54]
*SLC35A2* (NM_005660)	c.804dupA; p.(Pro269ThrfsTer24)	RB	mMCD/MOGHE	1	[40,54]
*SLC35A2* (NM_005660)	c.842G>A; p.(Gly281Asp)	RB	MOGHE	1	[41,54]
*SLC35A2* (NM_005660)	c.886_888delCTC; p.(Leu296del)	RB	mMCD/MOGHE	1	[40,54]
*SLC35A2* (NM_005660)	c.905C>T; p.(Ser302Phe)	RB	MOGHE	1	[54]
*SLC35A2* (NM_005660)	c.918_929delGCTGTCCACTGT; p.(Leu307_Val310del)	RB	MOGHE	1	[54]
*SLC35A2* (NM_005660)	p.(Cys210Tyr)	RB	MOGHE	1	[42]
*SLC35A2* (NM_005660)	p.(Pro15Thr)	RB	MOGHE	1	[42]
*SLC35A2* (NM_005660)	c.164G>T; p.(Arg55Leu)	RB	MCD	1	[56]
*SLC35A2* (NM_005660)	c.339_340insCTC; p.(Leu113dup)	RB	NLFE	1	[56]
*SLC35A2* (NM_005660)	c.747_757dup; p.(Ala253GlyfsTer100)	RB	MCD	1	[56]
*SLC35A2* (NM_005660)	c.910T>C; p.(Ser304Pro)	RB	NLFE	1	[56]
*TSC1* (NM_000368)	c.1525C>T; p.(Arg509Ter)	RB	FCD Iib	1	[41]
*TSC1* (NM_000368)	c.2074C>T; p.(Arg692Ter)	RB	FCD Iib	1	[41]
*TSC1* (NM_000368)	c.610C>T; p.(Arg204Cys)	RB	FCD Iia	1	[41]
*TSC1* (NM_000368)	c.64C>T; p.(Arg22Trp)	RB	FCD Iib	1	[58]
*TSC1* (NM_000368)	c.1741_1742delTT; p.(Phe581HisTer6)	CSF	FCD Iib	1	[33]
*TSC2* (NM_000548)	c.1372C>T; p.(Arg458Ter)	RB	FCD Iib	1	[41]
*TSC2* (NM_000548)	c.1754_1755delGT; p.(Tyr587Ter)	RB	HME	1	[39]
*TSC2* (NM_000548)	c.2251C>T; p.(Arg751Ter)	RB	FCD	1	[39]
*TSC2* (NM_000548)	c.2380C>T; p.(Gln794Ter)	RB	FCD Iib	1	[40]
*TSC2* (NM_000548)	c.4258_4261delCAGT; p.(Ser1420GlyfsTer55)	RB	FCD Iib	1	[58]
*TSC2* (NM_000548)	c.5228G>A; p.(Arg1743Gln)	RB	FCD Iib	1	[40]
*TSC2* (NM_000548)	c.3781G>A; p.(Ala1261Thr)	RB	FCD Iib	1 +	[34]
*TSC2* (NM_000548)	c.5227C>T; p.(Arg1743Trp)	RB	FCD Iib	1	[46]
WNT11	(chr11:64879188–135006516) LOH	RB	HHE	1	[35]
*ZNF337* (NM_001290261)	c.692_693del; p.(Thr231Arg fsTer45)	RB	FCD Iib	1 *	[46]

CNVs: copy number variants, CSF: cerebrospinal fluid, DNT: dysembryoplastic neuroepithelial tumor, FCD: focal cortical dysplasia, ffSWS: forme fruste of Sturge-Weber syndrome, GG: ganglioglioma, HHE: hypothalamic hamartoma epilepsy, HME: hemimegalencephaly, HS: hippocampal sclerosis, LGS: Lennox-Gastaut syndrome, LOH: loss-of-heterozygosity, MCD: malformations of cortical development; mMCD: mild MCD; MOGHE: mild malformation of cortical development with oligodendroglial hyperplasia in epilepsy, N: number of cases; NLFE: non-lesional focal epilepsy, RB: resected brain, SKS: Smith–Kingsmore syndrome, WS: West syndrome. +; * The same case with two variants.

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
