# Peer review of "CfDNA Measurement as a Diagnostic Tool for the Detection of Brain Somatic Mutations in Refractory Epilepsy"

_ijms, 2022, doi:10.3390/ijms23094879_

Round 1

Reviewer 1 Report

In the manuscript, the authors reviewed the literature about the possible use of cfDNA for diagnostic purposes in refractory epilepsy. Although the topic of the review is relevant and would be of interest to many researchers, the logic of the presentation and insufficient disclosure of many issues do not allow fully getting an idea of the possibility and purpose of using extracellular DNA for the diagnosis of refractory epilepsy. Below, major and minor issues are listed.

Major concerns

  1. Although the main idea of the manuscript is to consider cfDNA for diagnostic purposes, the description of methods that can be used to test cfDNA is almost absent.
  2. The section 2.3.3. “Treatment” is too short. There are many other treatment options including dietary ones. Another issue that should be disclosed in the review is the link between cfDNA and therapy.
  3. The section 2.1. “Somatic variants in the brain and refractory epilepsies” is hard to understand. Partially it repeats the information from the Table 1, however, some important ideas get lost among the long stream of numerical data.

Minor concerns

  1. In the title, the authors use “cfDNA as a diagnostic tool”, but cfDNA is a type of DNA but not a diagnostic tool. It should be rephrased.
  2. The authors discussed the detection of pathogenic CNVs, however, somatic copy number alterations (CNAs) are not even mentioned.
  3. There are several LOH variants in the Table 1, however, they are not discussed in the text, e.g., why they are pathogenic?
  4. Lines 272-275 “Although the difference was not statistically significant, more patients with mTOR pathway mutation responded to this treatment after three months of KD than patients without detectable mTOR pathway variants” is incorrect. There were only 12 and 13 patients in two groups. If the difference is not statistically significant, then the difference is absent, so this sentence should be rephrased or removed.
  5. Some other reviews on this topic should be mentioned in the Introduction.
  6. Figure 1 caption is too short. What do rectangles and ovals mean? Are all mTOR path members shown? Where are the genes, and where are the proteins, and where are the protein complexes? All abbreviations should be scheduled, e.g. what is GFR?
  7. For section titles 2.1 and 2.2, the terms “somatic variants” and “somatic alterations” are used, respectively. Do the authors imply any difference between them? Maybe it's worth revealing the terminology in the introduction.
  8. The subsection 2.2.1 “Cell-free DNA (brain tumors)” doesn’t correspond the subsection 2.2 “Somatic alterations in CSF cfDNA and refractory epilepsy”.
  9. Lines 215-217 “Until now, cfDNA from CSF is limited to targetted assays, and technical improvement is still required before it can be used for genetic screening in epileptic patients” - it is unclear why targeted methods cannot be used for screening. It seems that they are the ones that are more convenient to use than some bulky ones like WGS or WES. Or was there something else in mind?
  10. In the subsection 2.3.2. "Somatic brain variants detection in plasma cfDNA from patients with refractory epileptic" a lot is said about methylation and just about estimating the amount of cfDNA in plasma. This topic does not correspond to the subtitle.
  11. In the Abstract and Conclusion two similar phrases are seem to be incorrect or need more clarifications. Do genetic factors not always explain the etiology of refractory epilepsy?

“Although a genetic factor is estimated to occur in more than 70% of epileptic patients, the etiology of refractory epilepsy remains unknown in more than half of the cases.”

“Its etiology is unknown in approximately 60% of cases, although the existence of a genetic factor is estimated in about 75% of these individuals.”

  1. The authors say that cfDNA should be studied in plasma. But why is it bad or inconvenient to investigate it in the CSF? It is worth adding this comparing to the text of the manuscript.
  2. Lines 286-289 – «High throughput techniques might allow the screening for novel somatic mutation-specific for refractory epilepsy in plasma cfDNA, increasing the diagnostic yield of this disease with a minimally invasive procedure, enhancing the knowledge of its pathological mechanisms, and improving its treatment.» - the topic of high-throughput technique usefulness is missed in the manuscript. Therefore, it is unclear how such techniques can help in diagnostics and treatment. This discussion should be extended with accordance to Major concern #1.
  3. “Sim et al. (2018) also identified, in six patients with Lennox-Gastaut syndrome, different somatic pathogenic variants in SLC35A2” – check the grammar.
  4. Lines 195-197 “CSF is a source of circulating tumor DNA released upon tumor cell death and a potentially powerful biomarker for diagnosing and characterization tumors of the central 196 nervous system (CNS), such as gliomas” – check the grammar.
  5. Lines 203-205 “Although age and CSF collection in both groups were different, cfDNA concentration could be higher in epileptic patients than in controls,” the 1st part of the sentence is unclear.
  6. Line 208 – replace “essays” with “assays”.
  7. Lines 148-150 «They focused on 28 epilepsy-related genes, detecting 51 somatic variants of which, unless 26 were pathogenic or probably pathogenic, according to the American College of Medical Genetics» - check the grammar.
  8. There are no spaces between many words.

Reviewer 2 Report

1)The list of RE associated to somatic mosaic mutations is not exhaustive. I suggest to describe also other types of RE with a phenotype related to somatic mosaicism such as PCDH19 related epilepsy (OMIM#300088) or classic Rett Syndrome (Topçu M, Akyerli C, Sayi A, Törüner GA, KoçoÄŸlu SR, CimbiÅŸ M, Ozçelik T). Somatic mosaicism for a MECP2 mutation associated with classic Rett syndrome in a boy. Eur J Hum Genet. 2002 Jan;10(1):77-81. doi: 10.1038/sj.ejhg.5200745. PMID: 11896459; Pieras JI, Muñoz-Cabello B, Borrego S, Marcos I, Sanchez J, Madruga M, Antiñolo G. Somatic mosaicism for Y120X mutation in the MECP2 gene causes atypical Rett syndrome in a male. Brain Dev. 2011 Aug;33(7):608-11. doi: 10.1016/j.braindev.2010.09.012. Epub 2010 Oct 22. PMID: 20970936; Bourdon V, Philippe C, Bienvenu T, Koenig B, Tardieu M, Chelly J, Jonveaux P. Evidence of somatic mosaicism for a MECP2 mutation in females with Rett syndrome: diagnostic implications. J Med Genet. 2001 Dec;38(12):867-71. doi: 10.1136/jmg.38.12.867. PMID: 11768391; PMCID: PMC1734775.) or atypical Rett due to CDKL5 mutations (Kato T, Morisada N, Nagase H, Nishiyama M, Toyoshima D, Nakagawa T, Maruyama A, Fu XJ, Nozu K, Wada H, Takada S, Iijima K. Somatic mosaicism of a CDKL5 mutation identified by next-generation sequencing. Brain Dev. 2015 Oct;37(9):911-5. doi: 10.1016/j.braindev.2015.03.002. Epub 2015 Mar 27. PMID: 25819767). In this X-linked diseases, somatic mosaicism can bring to a phenotype also in males. This should be reported an explained in the review.

2)A table summarizing advantages and limits of NGS techniques and cf-DNA methodologies should be added to have a look at a glance of its employ in RE.

3) Pay attention to English, since several words are written attached, perhaps for formatting problems.

Reviewer 3 Report

Major issues

#1. It seems that there are many ungrammatical errors. Please use a professional native English proofreading service. At the round 2, the certification from the proofreading service company would be useful.

#2. Including the abbreviations, upper and lower cases, plural and single, ungrammatical errors etc. my impression of this review paper is roughly. Please consider a fine revision for the round2.

#3. The authors mentioned the clinical relationship between ID and ASD citing [13, 14]. However, clinical relationship among epilepsy and ASD or ID had not stated. Since this paper is about epilepsy, please cite some papers about the clinical relationship between epilepsy and ASD / ID such as epilepsy with FCD and autism, TSC with epilepsy and autism.

#4. The authors had stated the aim of the study in Introduction. However, this is a review article, but not a research article. The authors need to rephrase the part and should clearly state that this was a narrative type of review article.  Otherwise, the readers would expect the utility of cfDNA study and would be confused.

Minor issues

#1. AED is not currently recommended. Anti-seizure medication (ASM) is recommended.

#2. One the authors have used the abbreviation of RE, consistency is needed. Please see the last parts of Introduction.

#3. MTOR? mTOR?

#4. gDNA need the full words.

Round 2

Reviewer 2 Report

The explanations/modifications and this revised version of the paper fulfill my requirements.